# Determination of Heavy Metal Content: Arsenic, Cadmium, Mercury, and Lead in Cyano-Phycocyanin Isolated from the Cyanobacterial Biomass

**DOI:** 10.3390/plants12173150

**Published:** 2023-09-01

**Authors:** Daiva Galinytė, Gabrielė Balčiūnaitė-Murzienė, Jūratė Karosienė, Dmitrij Morudov, Rima Naginienė, Dalė Baranauskienė, Jurgita Šulinskienė, Ieva Kudlinskienė, Arūnas Savickas, Nijolė Savickienė

**Affiliations:** 1Department of Pharmacology, Faculty of Pharmacy, Academy of Medicine, Lithuanian University of Health Sciences, Sukileliu Av. 13, 50162 Kaunas, Lithuania; nijole.savickiene@lsmuni.lt; 2Faculty of Pharmacy, Institute of Pharmaceutical Technologies, Academy of Medicine, Lithuanian University of Health Sciences, Sukileliu Av. 13, 50162 Kaunas, Lithuania; gabriele.balciunaite@lsmuni.lt; 3Laboratory of Algology and Microbial Ecology, Nature Research Centre, Akademijos St. 2, 08412 Vilnius, Lithuania; jurate.karosiene@gamtc.lt (J.K.); dmitrij.morudov@gamtc.lt (D.M.); 4Laboratory of Toxicology, Neurosciences Institute, Academy of Medicine, Lithuanian University of Health Sciences, Eivenių Str. 4, 50161 Kaunas, Lithuania; rima.naginiene@lsmuni.lt (R.N.); dale.baranauskiene2@lsmu.lt (D.B.); jurgita.sulinskiene@lsmu.lt (J.Š.); 5JSC “Naujasis Nevėžis”, Jiesios St. 2, 53288 Ilgakiemis, Lithuania; ieva.kudlinskiene@nevezis.lt; 6Department of Drug Technology and Social Pharmacy, Academy of Medicine, Lithuanian University of Health Sciences, Sukileliu Av. 13, 50162 Kaunas, Lithuania; arunas.savickas@lsmu.lt

**Keywords:** cyano-phycocyanin, *Microcystis*, *Aphanizomenon*, *Spirulina platensis*, heavy metals

## Abstract

Cyano-phycocyanin (C-PC) is a light-absorbing biliprotein found in cyanobacteria, commonly known as blue-green algae. Due to its antioxidative, anti-inflammatory, and anticancer properties, this protein is a promising substance in medicine and pharmaceuticals. However, cyanobacteria tend to bind heavy metals from the environment, making it necessary to ensure the safety of C-PC for the development of pharmaceutical products, with C-PC isolated from naturally collected cyanobacterial biomass. This study aimed to determine the content of the most toxic heavy metals, arsenic (As), cadmium (Cd), mercury (Hg), and lead (Pb) in C-PC isolated from different cyanobacterial biomasses collected in the Kaunas Lagoon during 2019–2022, and compare them with the content of heavy metals in C-PC isolated from cultivated *Spirulina platensis* (*S. platensis*). Cyanobacteria of *Aphanizomenon flos-aquae* (*A. flos-aquae*) dominated the biomass collected in 2019, while the genus *Microcystis* dominated the biomasses collected in the years 2020 and 2022. Heavy metals were determined using inductively coupled plasma mass spectrometry (ICP-MS). ICP-MS analysis revealed higher levels of the most investigated heavy metals (Pb, Cd, and As) in C-PC isolated from the biomass with the dominant *Microcystis* spp. compared to C-PC isolated from the biomass with the predominant *A. flos-aquae*. Meanwhile, C-PC isolated from cultivated *S. platensis* exhibited lower concentrations of As and Pb than C-PC isolated from naturally collected cyanobacterial biomass.

## 1. Introduction

In recent decades, there has been an observed increase in the mass of cyanobacteria and the intensity of blooms in water bodies, influenced by eutrophication processes, the rising CO_2_ content in the atmosphere and surface waters, and global climate warming. The excessive presence of cyanobacteria in water bodies poses problems for the entire water ecosystem. One potential solution is the collection of cyanobacteria from water bodies and utilizing the collected biomass across various fields [1]. The aim of our research was to isolate C-PC from cyanobacterial biomass collected in nature and employ it in the production of pharmaceutical or cosmetic products. Ensuring the safety of the active ingredient is crucial during the development of such products. The concern is that cyanobacteria, like many other algae, possess a remarkable ability to sorb metal ions. Studies have shown that specific concentrations of heavy metals affect algal growth, bioactivity, and the process of photosynthesis. Indeed, the entire aquatic ecosystem is at risk when algae are exposed to high concentrations of heavy metals [2]. Cyanobacteria is even used as a bioindicator of water pollution due to its ability to accumulate contaminants. It has been observed that metals can bind to C-PC as well [3]. Generally, heavy metals are elements with a high atomic weight (between 63.5 g/mol and 200.6 g/mol) and a density exceeding 5.0 g/cm^3^. Therefore, out of 90 naturally occurring elements, 53 can be classified as heavy metals [4,5]. Metals are non-degradable, can accumulate in water and sediments, and persist in the environment for long periods of time, posing a threat to all living organisms when their concentrations exceed certain thresholds [6,7]. Certain heavy metals, such as As, Cd, Pb, and Hg, are particularly problematic due to their high levels of toxicity [4,8,9]. Heavy metals often contaminate surface water through the discharge of industrial or domestic wastewater disposed into sewers or water bodies [6]. Furthermore, pollutants can also be transported to water bodies from the soil through intensive agricultural practices or deposited from the atmosphere [2]. The high solubility of heavy metals in water makes them accessible to aquatic organisms, causing toxicity [10]. Therefore, there is a valid concern about the binding of heavy metals to C-PC. The level of C-PC contamination with metal ions depends on various factors, including the geographical location of the water body, the concentrations of heavy metals, the algae content in the water body, the water temperature, the pH, and the methods used for isolation and purification [6,10,11,12]. Consequently, the need for necessary monitoring and evaluation of heavy metals in C-PC using the latest research methods is crucial to ensure the safety aspects related to the content of heavy metals in raw materials of natural origin and to maintain their quality.

This study aimed to determine the concentrations of the most toxic heavy metals, namely lead (Pb), cadmium (Cd), arsenic (As), and mercury (Hg), in C-PC isolated from different biomasses of cyanobacteria collected in water bodies. The dominant genus and species of cyanobacteria depend on the geographical location, environmental and growth conditions [1]. For example, in the Kaunas Lagoon, where the biomass was collected for this study, cyanobacteria of *A. flos-aquae* dominated the biomass collected in 2019, while the genus *Microcystis* dominated the biomasses collected in the years 2020 and 2022. The content of heavy metals in C-PC isolated from the biomass of cyanobacteria collected in nature was compared with the concentrations of heavy metals in C-PC isolated from cultivated *S. platensis*. Purified C-PC is usually obtained from *S. platensis* by combining ammonium sulfate precipitation and different chromatographic methods [13]. Since cultivated *S. platensis* is protected as much as possible from the effects of heavy metals, C-PC of this origin was chosen as a comparison substance.

## 2. Results

To the best of our knowledge, this study represents the first investigation into the concentrations of heavy metals in C-PC isolated from cyanobacterial biomass collected in the Kaunas Lagoon, Lithuania. The results of the heavy metals analysis using ICP-MS on dry weight (dw) C-PC samples are presented in Figure 1. In the case of C-PC isolated from cyanobacterial biomass with dominant *A. flos-aquae* in 2019 (S1), the order of metal concentrations varied as follows: Pb 0.378 (0.013) μg/g > As 0.150 (0.012) μg/g > Hg 0.050 (0.006) μg/g > Cd 0.020 (0.000) μg/g (Figure 1a). The purity coefficient of C-PC (A620/A280) from sample S1 was 2.207 (0.015). For C-PC isolated from cyanobacterial biomass dominated by the genus *Microcystis* in 2020 (S2), the heavy metal concentrations ranged as follows: Pb 0.725 (0.006) μg/g, Hg 0.443 (0.007) μg/g, Cd 0.233 (0.005) μg/g, and As 0.190 (0.008) μg/g (Figure 1b). The determined purity coefficient of C-PC from sample S2 was 1.811 (0.028). In comparison with sample S2, the concentrations of heavy metals in C-PC from cyanobacterial biomass also dominated by the genus *Microcystis* but collected in 2022 (S3) varied in the following range: Pb 0.643 (0.015) μg/g > Cd 0.410 (0.014) μg/g > As 0.160 (0.008) μg/g > Hg 0.021 (0.003) μg/g (Figure 1c). The purity coefficient of C-PC from sample S3 was slightly lower than that of sample S2, reaching 1.463 (0.021). The concentrations of heavy metals in C-PC from cultivated *S. platensis* (S4) was also determined as follows: Hg 0.376 (0.016) μg/g > Cd 0.348 (0.005) μg/g > Pb 0.200 (0.008) μg/g > As 0.045 (0.006) μg/g (Figure 1d), and the purity coefficient of C-PC from sample S4 was identified as 1.130 (0.020). The purity coefficients of the C-PC samples are presented in Figure 2. The highest purity ratio was found in sample S1 and the lowest in sample S4, but both had lower amounts of the most tested heavy metals than samples S2 and S3.

As shown in Figure 1, C-PC extracted from cyanobacterial biomass collected in nature exhibited the highest levels of lead, particularly in samples S2 and S3 (*Microcystis*, 2020 and 2022), compared to the other tested elements. On the other hand, mercury and cadmium were the most predominant metals detected in the C-PC isolated from *S. platensis*.

### 2.1. Lead (Pb)

Figure 3a depicts the significant variation in lead levels among the different C-PC samples. The concentrations of lead ranged from 0.200 μg/g to 0.725 μg/g (dry weight), with the highest value observed in samples S2 and S3 (0.725 μg/g and 0.643 μg/g dw, respectively). These results indicate that C-PC isolated from the cyanobacterial biomass dominated by the genus *Microcystis* had the highest lead content. Sample S1 (*A. flos-aquae*, 2019) exhibited a concentration of lead that was more than 1.7 times lower, measuring 0.378 μg/g dw. The lowest concentration of lead (0.200 μg/g dw) was found in sample S4 (*S. platensis*). A significant difference in the lead content was observed among all four sample groups (*p* < 0.05). The order of the concentrations of Pb, as detected from the samples, was S2 > S3 > S1 > S4.

### 2.2. Cadmium (Cd)

The concentrations of Cd in the four C-PC samples ranged from 0.020 μg/g to 0.410 μg/g dw The highest Cd value (0.410 μg/g dw) was detected in sample S3 (*Microcystis*, 2022), followed by sample S4 (*S. platensis*)—0.348 μg/g dw and sample S2 (*Microcystis*, 2020)—0.233 μg/g dw. The lowest content of Cd (0.020 μg/g dw) was found in sample S1 (*A. flos-aquae*, 2019). Significant differences in Cd content were identified among all four sample groups (*p* < 0.05). The order of the concentrations of Cd was S3 > S4 > S2 > S1 (Figure 3b).

### 2.3. Arsenic (As)

No significant difference in As content was observed between samples S1 (0.150 μg/g dw) and S3 (0.160 μg/g dw), where C-PC was isolated from the biomass of *A. flos-aquae* and *Microcystis* spp. (2022), respectively. However, a statistically higher concentration of As was detected in C-PC from *Microcystis* biomass collected in 2020 (0.190 μg/g dw). The lowest value of As (0.045 μg/g dw) was determined in sample S4 (*S. platensis*). The order of the concentrations of arsenic observed among the samples was S2 > S3 > S1 > S4 (Figure 3c).

### 2.4. Mercury (Hg)

The highest concentration of Hg was determined in sample S2 (0.443 μg/g dw) followed by sample S4 (0.376 μg/g dw). A more than seven-fold lower concentration of Hg was found in sample S1 (0.050 μg/g dw), and the lowest Hg content was detected in sample S3 (0.021 μg/g dw). The results regarding the concentrations of mercury in the tested samples are controversial, since the highest and lowest Hg content was observed in the C-PC isolated from *Microcystis* biomass. Nevertheless, a significant difference in Hg content was observed among all four groups (*p* < 0.05). The order of the concentrations of Hg, from highest to lowest, as determined from the C-PC samples, was S2 > S4 > S1 > S3 (Figure 3d).

The results of ICP-MS analysis revealed higher concentrations of the tested metals in C-PC isolated from the cyanobacterial biomass collected from nature, particularly when the genus *Microcystis* dominated (in the years 2020 and 2022). On the other hand, samples S1 (*A. flos-aquae*, 2019) and S4 (*S. platensis*) exhibited lower concentrations of the tested metals. Cultivated *S. platensis* is more protected from heavy metal exposure from the environment. Meanwhile, the metals from natural and anthropogenic sources are released into aquatic systems [6]. The dominant genus of cyanobacteria, and the overall metal content in the aquatic system, can influence the binding of metals via C-PC. In addition, the purification processes of C-PC are also important. Gel filtration chromatography was utilized for purifying C-PC from *Microcystis* spp. (samples S2 and S3), resulting in purity coefficients of 1.811 (0.028) and 1.463 (0.021), respectively. Ion exchange chromatography was employed to isolate C-PC from *A. flos-aquae* (sample S1), with the resulting purity coefficient of 2.207 (0.015).

## 3. Discussion

### 3.1. Heavy Metal Content in the Water Body

The accumulation of metals in cyanobacteria is influenced by the metal concentrations in the water body. In this study, the cyanobacterial biomass was collected from the Kaunas Lagoon (Lithuania), the largest artificial water body in Lithuania. The Kaunas Lagoon was created in 1959 after the construction of the Kruonis hydro accumulative power plant and the damming of the Nemunas river—the longest river in Lithuania. As a result, the Kaunas Lagoon belongs to the Nemunas basin and is classified as a highly altered water body. In addition, it is considered at risk due to sewage, agriculture, and historical pollution [14]. Heavy metal monitoring in surface water has been conducted in Lithuania since 1993. According to the State Environmental Monitoring Program for 2018–2023, approved by the Government of the Republic of Lithuania on 3 October 2018, by resolution no. 996 “On the approval of the State Environmental Monitoring Program for 2018–2023”, the monitoring of metals in water, bottom sediments, and biota in the Kaunas Lagoon is carried out every three years. Bottom sediments are tested for metals that tend to accumulate in sediments, while the biota is tested for those metals that have established environmental quality standards, such as mercury (Hg). Therefore, we only have the results of the heavy metals in the Kaunas Lagoon from the year 2021 (reflecting the period from 2019 to 2022). To assess the pollution of the Kaunas Lagoon with heavy metals, we analyzed ten years of monitoring results. We considered the content of heavy metals (Pb, Cd, As, and Hg) in the surface water and bottom sediments of the Kaunas Lagoon from 2013 to 2022 based on the monitoring procedures of lakes and ponds in the country (Figure 4). The monitoring results revealed that the concentration of Cd in the surface water of the Kaunas Lagoon remained unchanged over the ten-year period (0.05 μg/L). Nonetheless, the concentration of cadmium increased in the bottom sediments. In the C-PC samples, a statistically significant increase in the concentration of Cd was observed from 0.020 μg/g (sample S1, 2019) to 0.410 μg/g (sample S3, 2022). The concentration of lead in the bottom sediments also increased from 14 mg/kg in 2015 to 28 mg/kg in 2021. In the C-PC samples, there was an almost two-fold increase in the concentration of Pb in 2020 (sample S2) compared to 2019 (sample S1) (*p* < 0.05). In 2022 (sample S3), the concentration of Pb in C-PC was 11% lower than in 2020 (sample S2), but still 70% higher than in 2019 (sample S1). The significantly lower concentrations of Cd and Pb in sample S1 (*Aphanizomenon*, 2019) could be attributed to the differences in the C-PC purification methods or the dominant algal genus. The concentration of As in the surface water remained relatively constant over the entire ten-year period, with no statistically significant differences observed between the C-PC samples from 2019 and 2022 (*p* > 0.05). The mercury concentration in the surface water and sediments remained relatively stable over the ten years. In 2021, the concentrations of hazardous substances in the water of the Kaunas Lagoon did not exceed the environmental quality standards based on both the maximum allowed concentrations and the annual average standards. However, in the biota of the Kaunas lagoon, the concentration of mercury (119 μg/kg) exceeded the maximum allowed concentration six times (20 μg/kg) [15].

After evaluating the condition of the water bodies in Lithuania in 2022, it was determined that 57% of rivers and 62% of lakes within the Nemunas basin area did not meet the criteria for good condition. The majority of water bodies failing to meet these requirements are located in the regions with intensive agriculture and farming. Agricultural activities, sewage discharge, hydromorphological effects, natural processes, and changing climatic conditions are the primary sources of water pollution in Lithuania. These factors contribute to the classification of the Kaunas Lagoon as a risk water body. Nonetheless, it is encouraging that more and more attention is being paid to the state of the water bodies in Lithuania; the reliability of these assessments has increased, and the states of the water bodies are not deteriorating. However, it has been acknowledged that it will take a considerable amount of time for water condition indicators to recover, even with the implementation of the necessary improvement measures. One such measure to achieve a good water body condition involves the removal of excess aquatic vegetation [16].

Overall, heavy metal contamination in the surface water is a global issue, and is particularly prevalent in Africa and Asia, where the highest concentrations of heavy metals are found. Lower concentrations are observed in European surface water [3].

The water temperature and pH levels can also influence the binding of heavy metals. The average annual water temperature in the Kaunas Lagoon decreased from 14.89 °C in 2019 to 14.01 °C in 2022, while the pH in 2019–2021 remained at 8.5, and slightly decreased in 2022 to 8.4 (Figure 5) [15]. However, detailed studies are required to assess whether these slight changes in the temperature and pH in the water of the Kaunas Lagoon could have influenced the accumulation of heavy metals in the cyanobacterial biomass.

Clearly, our water bodies are susceptible to heavy metal pollution. Cyanobacteria, including phycocyanin, have a tendency to bind heavy metals from the surrounding environment.

### 3.2. Aspects of Heavy Metal Binding by Cyanobacteria and C-PC

Cyanobacteria have a remarkable ability to bind metals from the surrounding environment. It is known that dead algal cells can absorb more metals than living ones. The term “bio-sorption” is typically used to describe the sorption of metals by dead biomass, while “accumulation” refers to the living cells. Two distinct processes affect the accumulation of heavy metals in cyanobacteria: passive initial uptake and active uptake. The passive initial uptake process does not depend on cell metabolism and lasts a few seconds or minutes. During this time, metal ions are adsorbed onto the cell surface. In water, metals are predominantly found in their cationic form. The cell wall of cyanobacteria contains various functional groups, such as hydroxyl (–${OH}^{-}$), carboxyl (–${COOH}^{-}$), sulfhydryl (–${SH}^{-}$), amino (–${NH}_{2}^{-}$), and phosphoryl (${PO}_{3}^{-3}$), which are associated with cell wall components, among which include the following: teichuronic acid, teichoic acids, peptidoglycans, polysaccharides, and proteins. These functional groups provide a negative charge to the cell surface, allowing metal ions to be adsorbed on the cell surface due to electrostatic interactions. The dissociation constant, pKa, is specific to each functional group and determines the dissociation into the corresponding anions and protons at a particular pH. This supports the fact that changes in the pH of the water body affect the binding of heavy metals. The second process is much slower and depends on metabolism; metal ions pass through the cell membrane into the cytoplasm [6,11]. The cell membrane acts as a barrier and prevents unwanted substances from entering the cell. However, different transport proteins, responsible for nutrient absorption, assist heavy metals in crossing the membrane [4]. For instance, As^5+^ is taken up via phosphate transporters, while As^2+^ uses glycerol transport proteins [17]. Pb^2+^ and Cd^2+^ enter the cell through divalent metal transport channels and carrier proteins, including divalent cation transporter 1 (DCT1) [4]. Both organic and inorganic forms of mercury compounds can be transported via the specific transport proteins MerC and MerT [18].

The capacity for metal sorption may vary among the genera of cyanobacteria due to the differences in their cell component distributions and the types of functional groups present. *Microcystis aeruginosa* and *Microcystis flos-aquae* form multicellular aggregates surrounded by a colonial capsule or an exopolymeric matrix. A strong interaction between the mucus layer from *Microcystis* spp. and cations has been shown. The several functional groups, especially the carboxyl groups, arranged in the cell walls of *Microcystis* spp. provide biosorption sites for metal binding and lead to a high capacity for binding heavy metals [11]. While the genus *Aphanizomenon* is planktonic cyanobacteria with straight trichomes, it can exist as single filaments or form aggregated trichomes in parallel fascicles, reaching a size of up to 2 cm [19]. It has been demonstrated that *Microcystis aeruginosa* exhibits a high affinity for binding Hg^2+^, Cd^2+^, and Pb^2+^ [20]. A more elevated cell surface influences the more remarkable ability of *Microcystis aeruginosa* to absorb Hg^2+^ than *A. flos-aquae* [21]. Our study also supports the high affinity of *Microcystis* for Hg, Cd, and Pb ions.

*S. platensis* is a filamentous, non-differentiated, spiral-shaped, multicellular cyanobacteria [22]. Gelagutashvili E et al. reported that *S. platensis* has a higher sorption capacity for cadmium [23]. However, metals, such as lead, mercury, cadmium, and arsenic, can also be found in *Spirulina* products [22]. In our study, C-PC isolated from *S. platensis* predominantly contained Hg and Cg among the metals examined.

C-phycocyanin is a light-harvesting phycobiliprotein that plays an important role in phycobilisomes, which are attached to the outer membrane of the chloroplast thylakoid [24,25]. Phycobiliproteins are oligomers, with the hexamer being the basic building block [26]. Each monomer of C-PC consists of α- and β-polypeptide chains with three covalently binding open-chain tetrapyrrole chromophores, known as phycocyanobilins. This binding occurs through a thioether bond between a cysteine residue and the A ring of tetrapyrrole. The α-subunit of C-PC contains a phycocyanobilin covalently attached to α-84 cysteine, while the β-subunit contains two phycocyanobilins that are linked to the β-84 and β-155 cysteine residues of the apoprotein, respectively [26]. The mechanisms underlying heavy metal binding to C-phycocyanin may vary with the different metal cations, but they are not fully understood [12]. Gelagutashvili E. and others distinguished two feasible ways of heavy metal ion binding to C-PC: site-specific and non-specific (diffuse) binding. Hexamers of C-PC have a globular structure, and the negative electrostatic field is dominant. Therefore, cations of heavy metals can be electrostatically attached to the strong anionic area at the center of the hexamer. This interaction is non-specific, while the binding of the metal ions form site-specific bonds to the thiol groups of the protein. Gelagutashvili E. and others’ research suggested that cysteine at the β-84 position is one of the specific binding sites for metal ions in C-PC [23]. Heavy metals can affect the conformational changes in C-PC and inhibit the photosynthesis processes of cyanobacteria [3,4]. Higher concentrations of toxic metals lead to more changes in the structure of the C-PC macromolecule [3]. Pb^2+^ and Cd^2+^ form aggregations with C-PC molecules and modify their secondary structure (α helix). Research has shown that the effect of Pb^2+^ on the Tyr residues of PC was more significant than the Trp residues, while the effect of Cd^2+^ was more significant on the Trp residue of PC. The changes in the protein skeleton are possibly related to the loss of C-PC function [27,28].

The binding affinities of the heavy metal ions depend on the specific metal. The binding constants are arranged in descending order, as follows: Hg > Ag > Pb > Cr > Cu [23]. In addition, the binding affinities of the heavy metals depend on the metal concentrations, and that is why the research results may vary. The binding of the heavy metals induces conformational changes in the C-PC molecules and fluorescence quenching. For instance, Chi Z. and others referred to the fluorescence quenching effect on the C-PC with the order of Hg^2+^ > Cu^2+^ ≈ Ag^+^ > Pb^2+^ > Cr^3+^ > Zn^2+^ at a concentration of 50 μM, while the lower concentrations changed the order to Hg^2+^ > Cu^2+^ > Ag^+^ > Zn^2+^ > Cd^2+^ > Pb^2+^. Understanding the influence of heavy metal ions on C-PC at low concentrations is essential for assessing the toxicity of accumulated heavy metal ions in aquatic systems. Studies conducted at lower concentrations may be more important for evaluating the mechanisms of heavy metal binding to C-PC and their impact of toxicity on aquatic systems since they better reflect natural conditions. However, it should be noted that the highest concentrations of heavy metals result in more changes in the structure of the C-PC macromolecule [3].

The findings of our study revealed that lower concentrations of heavy metals were found in C-PC isolated from the cyanobacterial biomass with the dominant *A. flos-aquae* compared to the C-PC samples isolated from the biomass with the dominant *Microcystis* spp. The mucus layer and the larger cell surface area of *Microcystis* spp. may have influenced these results.

### 3.3. The Influence of the Purification Method on the Heavy Metal Content in C-PC

C-PC isolation from cyanobacterial biomass encompasses different steps, such as cell disruption, primary extraction, and purification [29]. Our study conducted gel filtration chromatography on a Sephadex G-25 column to purify C-PC from biomass with dominant *Microcystis* spp. Meanwhile, ion exchange chromatography on a Q-Sepharose XL column was used to purify C-PC from biomass with predominant *A. flos-aquae*. Both methods are well-documented and commonly employed for C-PC purification. Purification methods must protect the product from degradation (as C-PC is known to be sensitive to pH and temperature changes), help obtain the highest possible purity level, and maintain cost-effectiveness [29]. The combination of these two methods is also often used for C-PC purification [30]. The attraction of oppositely charged molecules supports the ion exchange chromatography of proteins. The protein’s surface can be charged depending on the isoelectric point and the pH of the environment. The ion exchanger usually has a base matrix with porous beads that provide a sufficient adsorption surface. Positively or negatively charged ligands are immobilized on this matrix. Cation and anion exchangers are distinguished, which are negatively and positively charged, respectively. The protein is negatively charged above its isoelectric point and binds to an anion exchanger. On the contrary, the protein is positively charged below its isoelectric point and binds to cation exchangers [31]. The significant advantage of ion exchange chromatography is that it ensures mild separation conditions that enable the proteins from conformational changes [32]. C-PC should be purified depending on the field of use [33]. The purity of C-PC is typically estimated according to the absorbance ratio of A620/A280, wherein a purity of 0.7 is considered a food grade, 3.9 as a reactive grade, and values greater than 4.0 as an analytical grade [34].

The purity grades of the C-PC samples S1, S2, S3, and S4 were 2.207 (0.015), 1.811 (0.028), 1.463 (0.021), and 1.130 (0.020), respectively (Figure 2). However, the purity coefficients of C-PC do not reflect heavy metal contamination (Figure 1) in our study.

An anionic Q-Sepharose XL matrix used for ion exchange chromatography could influence the lower concentrations of heavy metals in C-PC isolated from cyanobacterial biomass with the dominant *A. flos-aquae*. However, further detailed studies are required to confirm this hypothesis. In addition, it is important to note that our study did not aim to find correlations between these purification methods. Metal content was determined to evaluate the safety of the isolated phycocyanin for further research.

### 3.4. Regulation of Heavy Metal Content in Products

Since heavy metals are naturally present in the environment, they can also be present as traces in food products, medical devices, or pharmaceuticals. Heavy metals can enter the final product with active substances or excipients through contaminated water, manufacturing equipment, or container systems. Phycocyanin is used as a natural blue dye in food (chewing gum, dairy products, drinks, desserts, etc.) and cosmetics (e.g., lipsticks) [33,35]. C-PC also demonstrates numerous medical applications due to its antioxidant, anti-inflammatory, and antitumor properties [36,37]. Therefore, the possible contamination of C-PC with heavy metals should be carefully monitored to ensure the quality and safety of the final product.

#### 3.4.1. Food Products

Heavy metal content in food must be maintained as low as can be reasonably achieved following good working practices [38]. The maximum levels for specific contaminants in foodstuff, including lead, cadmium, and mercury, are set under the Commission Regulation (EU) 2023/915 for the protection of public health. The maximum levels of lead and mercury in food supplements is set at 3 mg/kg and 0.10 mg/kg, respectively. The maximum level of cadmium in food supplements is established as 1 mg/kg, except for food supplements exclusively or mainly consisting of dried seaweed, products derived from seaweed, or dried bivalve mollusks—3 mg/kg. The maximum levels of arsenic are set only for infant or baby products and, depending on the product, should not exceed 0.010–0.020 mg/kg. The maximum levels of As in food supplements have not been identified [39]. There is limited data on the heavy metal content of C-PC, but studies on the determination of heavy metals in algae products were found. Al-Dhabi N.A. evaluated the content of six heavy metals, Ni, Zn, Hg, Pt, Mg, and Mn, in *Spirulina* food products for human consumption. All tested metals were within the daily intake levels. The content of Hg ranged from 0.002 mg/kg dw to 0.028 mg/kg dw [22]. Sandaruber F. and others determined heavy metal concentrations in 15 commercially available microalgae, including *S. platensis* and *A. flos*-*aquae* powders. The concentrations of heavy metals (means and standard deviations) varied within the following wide limits: from 0.89 (0.14) μg/100 g to 799.2 (14.8) μg/100 g in *S. platensis* and 60.37 (0.05) μg/100 g in *A. flos-aquae* for As, from 0.54 (0.26) μg/100 g to 9.61 (1.56) μg/100 g in *S. platensis* and 0.58 (0.30) μg/100 g in *A. flos-aquae* for Cd, from 0.25 (0.07) μg/100 g to 4.91 (1.02) μg/100 g in *S. platensis* and 0.3835 (0.0001) μg/100 g in *A. flos-aquae* for Hg, and from 4.25 (5.31) μg/100 g to 22.7 (16.06) μg/100 g in *S. platensis* and 4.98 (6.33) μg/100 g in *A. flos-aquae* for Pb. The concentrations of Cd, Hg, and Pb in the tested *S. platensis* and *A. flos*-*aquae* products did not exceed the maximum levels specified in Regulation EU 2023/015. The highest concentrations of Hg and As among all the different tested algal powders were determined in *S. platensis*. However, one product from *S. platensis* had the lowest concentrations of all the four determined heavy metals combined. Notably, the highest concentrations of heavy metals were found in products from China, where the regulations for heavy metals were only set for macroalgae but not microalgae. Therefore, the legislation on food supplements varies between the EU, the US, and Asian countries. Even in the cultivation of algae, an impure cultivation medium can cause the contamination with heavy metals [40].

#### 3.4.2. Cosmetic Products

The manufacturing of cosmetic products must also adhere to good manufacturing practices. According to Regulation (EC) No. 1223/2009, heavy metals, such as arsenic, cadmium, lead, and mercury (with some exceptions), and their compounds are included in the list of substances prohibited in cosmetic products. If preservatives, such as thiomersal or phenylmercuric salts, are used in cosmetic products, the maximum allowable mercury concentration is 0.007% of the ready-to-use preparation [41]. If the cosmetic product contains traces of prohibited substances, the evidence must be provided that such traces were technically unavoidable. The safety assessor must decide whether the trace level is toxicologically acceptable and whether the product is safe [42].

#### 3.4.3. Medical Devices

Arsenic, lead, cadmium, mercury, and their compounds are included in the list of substances that are carcinogenic, mutagenic, or toxic to reproduction (CMR). Thus, according to Regulation 2017/745, the concentration of these substances in medical devices must not exceed 0.1% by mass (*w*/*w*), with some exceptions provided in this Regulation when a higher amount of such substances must be justified. It is important to consider whether the intended device will be used to treat children, pregnant or lactating women, or other patient groups considered particularly vulnerable to such agents [43,44].

#### 3.4.4. Medicines

The European Medicines Agency categorizes the elements Cd, As, Pb, and Hg as Class 1 substances based on their toxicity to humans. They should not be used in the manufacturing of pharmaceuticals, and their usage must be strictly limited. The ICH guidelines on elemental impurities establishes the permitted daily exposure (PDE) of each element of toxicological concern. The oxidation state of the element in the drug product, the route of administration, relevant animal and human studies, and safety data are the factors considered in the safety assessment of these elements. Established PDE and recalculated permitted concentrations of these elemental impurities in drug products, drug substances, and excipients with daily doses of not more than 10 g per day for oral, parental, inhalation, and cutaneous exposure are provided in Table 1. This guideline applies to newly developed drug products, including products with proteins and polypeptides, but does not apply to herbal products [45].

Therefore, the purification process is critically important for ensuring the safety of the natural source material. In this study, our aim was to assess the residual heavy metal content after the purification of C-PC. According to the determined content of heavy metals, C-PC isolated from the cyanobacterial biomasses collected in nature could be used in the development of cosmetics, medical devices, or pharmaceutical products. However, other ingredients may also contain heavy metal impurities and heavy metals may enter the product during manufacturing. Thus, it is essential to consider the permitted concentrations of these heavy metals in the final product.

## 4. Materials and Methods

### 4.1. Collection of Cyanobacterial Biomass

The biomass of cyanobacteria was collected with a plankton net (with a mesh size of 20 μm) from the Kaunas Lagoon during cyanobacteria bloom in September–October 2019, 2020, and 2022. The collected biomass was frozen and stored at −20 °C in a freezer. In addition, phytoplankton samples were taken from the surface water layer (0.10–0.30 m) for microscopic analyses using a Ruttner water sampler to examine the bloom-forming species. Samples were preserved with 4% (*v*/*v*) formaldehyde solution. Microscopic analysis revealed that species of the genus *Aphanizomenon,* especially *A. flos-aquae,* dominated in the year 2019 (constituting up to 99% of the total phytoplankton biomass), while species of the genus *Microcystis* prevailed in the years 2020 and 2022 (up to 98%).

### 4.2. C-PC Extraction and Purification

Phycocyanin extraction and purification were conducted using wild cyanobacterial biomass following the method described by Khazi et al. (2018), with some modifications [46].

C-PC was extracted using one freeze-thaw cycle (freezing at −20 °C and thawing at 20 ± 2 °C). The supernatant for purity was collected after centrifugation at 8000× *g* for 10 min and precipitated with a 50% saturated ammonium sulfate solution overnight at +4 °C. The ammonium sulfate-precipitated solution was centrifuged at 10,000× *g* for 10 min and the precipitate was resuspended with 10 mM Na-phosphate buffer (pH 7.0). Desalting was performed via diafiltration using a membrane with a pore size of 10 kDa. Gel filtration chromatography on a Sephadex G-25 column was used to purify C-PC from biomass with dominant *Microcystis* spp. The column was pre-equilibrated and eluted with 10 mM Na-phosphate buffer (pH 7.0), with the flow rate set at 0.5 mL/min. Purification of C-PC from the biomass with dominant *A. flos-aquae* was carried out through ion exchange chromatography on a Q-Sepharose XL column. After pre-equilibration of the column with 10 mM Na-phosphate buffer (pH 7.0), elution was performed using a linear gradient with 0.5 M NaCl and the same buffer at a flow rate of 0.5 mL/min. The purified C-PC extract was freeze-dried after desalting.

### 4.3. Materials: C-PC Partially Purified from S. platensis

C-PC partially purified from *S. platensis* was purchased from Sigma-Aldrich (St. Louis, MO, USA).

### 4.4. Purity Assessment of Phycocyanin

The purity of C-PC was determined spectrophotometrically, according to the following formula (Bennett, Bogodar, 1973) [47]:(1)C-PC purity=OD620/OD280,
where OD280 indicates the absorbance of total proteins and OD620 indicates the absorbance of phycocyanin. The experiment was repeated three times.

### 4.5. C-PC Sample Collection

Heavy metals were determined in four lyophilized phycocyanin samples, summarized in Table 2. The first sample (S1) of C-PC was isolated from cyanobacterial biomass collected in the Kaunas Lagoon (Lithuania) in 2019. *A. flos-aquae* was dominated in this biomass. Ion exchange chromatography was used to purify C-PC. The second (S2) and the third (S3) samples were isolated from cyanobacterial biomass collected in the Kaunas Lagoon in the years 2020 and 2022, respectively. The genus of *Microcystis* dominated these biomasses. Gel filtration chromatography was used for the purification of C-PC. The fourth sample (S4) was C-PC isolated from cultivated *S. platensis* (Sigma-Aldrich). This sample was taken as a reference for C-PC isolated from naturally collected cyanobacterial biomass.

### 4.6. Inductively Coupled Plasma-Mass Spectrometry

The concentrations of the heavy metals—lead (Pb), cadmium (Cd), arsenic (As), chromium (Cr), and mercury (Hg)—were analyzed using an inductively coupled plasma mass spectrometer NexION™ 300 D (PerkinElmer, Shelton, CT, USA) equipped with nickel cones and a quartz cyclonic spray chamber as a sample introduction system and using the standard mode (STD). The optimized instrument conditions and measurement parameters are listed in Table 3.

The calibration solutions for ICP-MS analysis were prepared using the Pure Plus 10 mg/L Multi-Element calibration standard 3 (PerkinElmer, Shelton, CT, USA) and the 10 mg/L Hg (PerkinElmer, Shelton, CT, USA) calibration standard. Germanium (Ge) was added as an internal standard to the samples.

Calibration graphs for the determinations of lead, cadmium, arsenic, and mercury in samples using ICP-MS were prepared in the concentration range from 0 to 2 µg/L. Correlation coefficients for lead, cadmium, arsenic, and mercury were 0.9999, 0.9999, 0.9999, and 0.9986, respectively. Data of the limits of detection (LOD) and limits of quantification (LOQ) for these four trace elements are presented in Table 4.

To determine trace elements—Pb, Cd, As, Cr, and Hg—a closed quartz vessel and the microwave oven Multiwave 3000 (Anton Paar GmbH, Graz, Austria) digestion procedure were employed. A 100 mg of C-PC sample was accurately weighed into quartz digestion vessels and 1 mL of 65% HNO_3_, and 1 mL of H_2_O_2_ were added to the digestion vessels and digested through following the microwave program: 60 bar pressure, 200 °C temperature, and 800 W microwave power. The digests were transferred into pre-cleaned polyethylene containers and diluted with ultra-purified water. Heavy metal concentrations were expressed as micrograms per gram of C-PC on a dry weight basis. The experiment was repeated four times.

### 4.7. Statistical Analysis

These data were statistically calculated and analyzed using SPSS V29.0. Metal concentrations and purity coefficients of the C-PC samples were expressed as means. The one-way ANOVA and Bonferroni test (with equal variances assumed) was employed to determine the statistical significance (*p* < 0.05) of concentrations of the individual elements among the C-PC samples and purity coefficients of the C-PC samples, while Dunnett’s T3 test (with equal variances not assumed) was employed to determine the statistical significances (*p* < 0.05) of different metal concentrations in the same C-PC sample.

## 5. Conclusions

ICP-MS analysis of heavy metals revealed higher levels of the most investigated heavy metals in C-PC isolated from the biomass with the dominant *Microcystis* spp. compared to C-PC isolated from the biomass with the predominant *A. flos-aquae*. Additionally, C-PC isolated from cultivated *S. platensis* exhibited lower concentrations of As and Pb compared to C-PC isolated from naturally collected cyanobacterial biomass. However, it had significant concentrations of Cd and Hg. These results could have been influenced by several factors, such as the dominant algal species, which, due to their structural features, have different capacities to attract heavy metals, as well as the purification method and the purity grade of C-PC.

## Figures and Tables

**Figure 1 plants-12-03150-f001:**
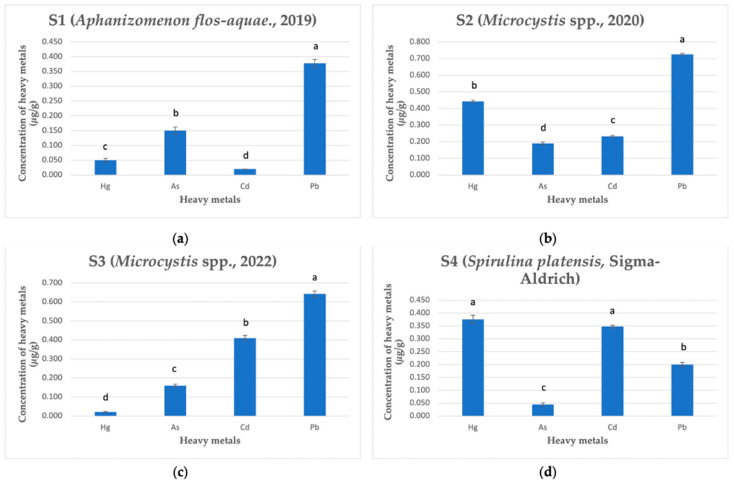
Content of heavy metals in C-PC samples (μg/g dw). (**a**) Heavy metal concentrations in sample S1. (**b**) Heavy metal concentrations in sample S2. (**c**) Heavy metal concentrations in sample S3. (**d**) Heavy metal concentrations in sample S4. Different letters above the bars indicate significant differences among the different metal concentrations (*p* < 0.05) identified using Dunnett’s T3 test. Means and standard deviations are presented. The experiment was repeated 4 times.

**Figure 2 plants-12-03150-f002:**
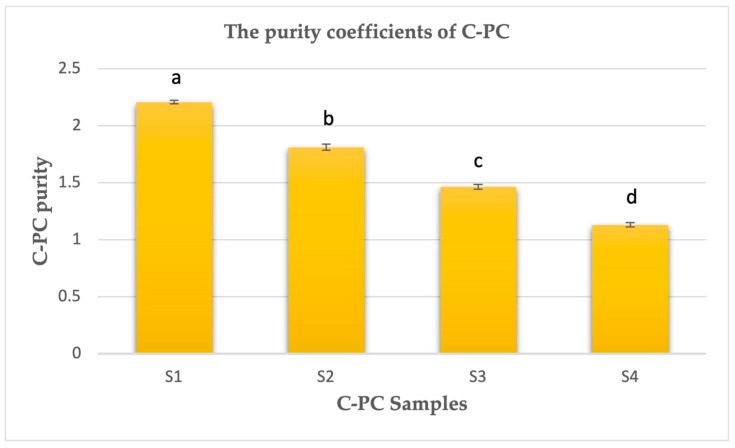
The purity coefficients of C-PC samples S1, S2, S3, and S4. Different letters above the bars indicate significant differences (*p* < 0.05) identified using Bonferroni’s test. Means and standard deviations are presented. The experiment was repeated 3 times.

**Figure 3 plants-12-03150-f003:**
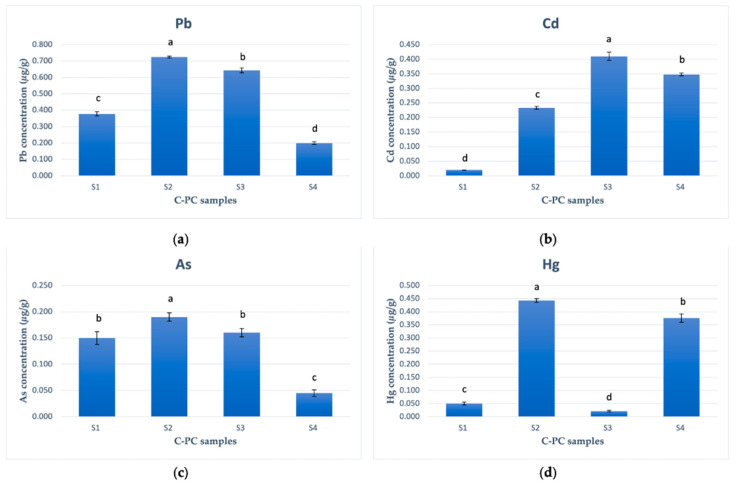
Graphical representation of heavy metal concentrations (μg/g dw) in C-PC samples. (**a**) Pb concentration. (**b**) Cd concentration. (**c**) As concentration. (**d**) Hg concentration. Different letters above the bars indicate significant differences among the different metals (*p* < 0.05) identified using Bonferroni’s test. Means and standard deviations are presented. The experiment was repeated 4 times.

**Figure 4 plants-12-03150-f004:**
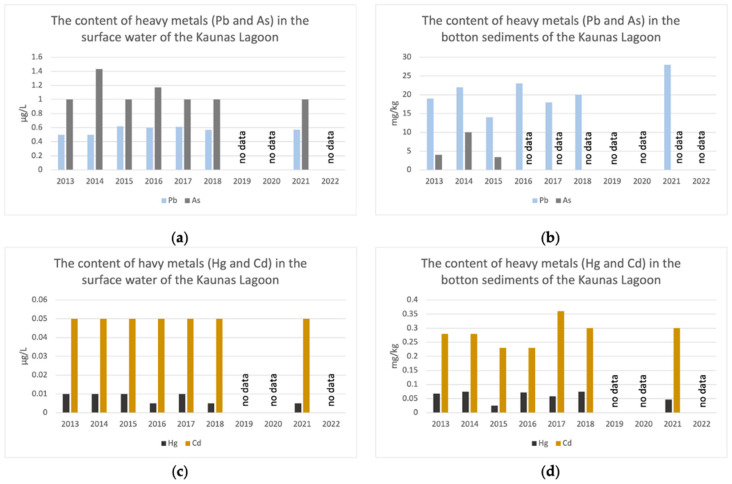
Heavy metal content (μg/L) in the surface water and bottom sediments of the Kaunas Lagoon. (**a**) Pb and As content in the surface water of the Kaunas Lagoon. (**b**) Pb and As content in the bottom sediments of the Kaunas Lagoon. (**c**) Hg and Cd content in the surface water of the Kaunas Lagoon. (**d**) Hg and Cd content in the bottom sediments of the Kaunas Lagoon [15].

**Figure 5 plants-12-03150-f005:**
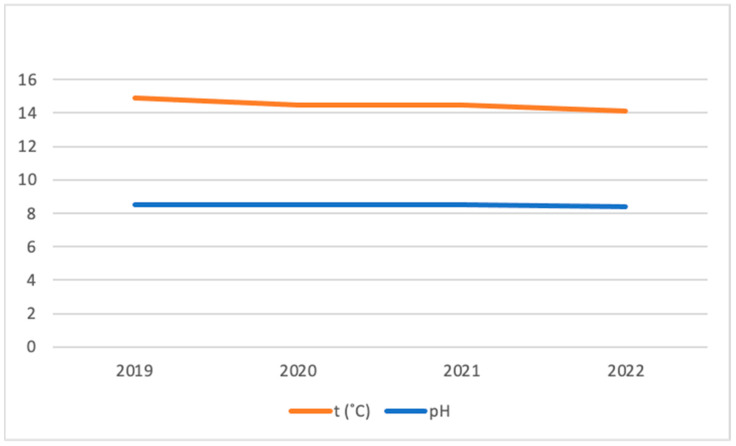
Changes in the average annual water temperature and pH of the Kaunas Lagoon in 2019–2022.

**Table 1 plants-12-03150-t001:** Maximum allowed levels for heavy metals in food supplements, cosmetic products, medical devices, and medicines.

Products	Maximum Levels for Heavy Metals				References
	Pb	Cd	As	Hg	
Food supplements	3 mg/kg	1 mg/kg (3 mg/kg) ^1^	Not identified	0.10 mg/kg	Regulation (EU) 2023/915 [38]
Cosmetic products	Prohibited ^2^	Prohibited ^2^	Prohibited ^2^	0.007% of ready-to-use preparation ^3^	Regulation 1223/2009 [41]
Medical devices	≤0.1% by mass (*w*/*w*) ^4^	≤0.1% by mass (*w*/*w*) ^4^	≤0.1% by mass (*w*/*w*) ^4^	≤0.1% by mass (*w*/*w*) ^4^	Regulation 2017/745 [42]
Medicines ^5^					
Oral	5 μg/day (0.5 μg/g)	5 μg/day (0.5 μg/g)	15 μg/day (1.5 μg/g)	30 μg/day (3 μg/g)	ICH guideline Q3D (R2) on elemental impurities [44]
Parental	5 μg/day (0.5 μg/g)	1.7 μg/day (0.2 μg/g)	15 μg/day (1.5 μg/g)	3 μg/day (0.3 μg/g)	
Inhalation	50 μg/day (5 μg/g)	3 μg/day (0.3 μg/g),	1.9 μg/day (0.2 μg/g)	1.2 μg/day (0.1 μg/g),	
Cutaneous	50 μg/day (5 μg/g)	20 μg/day (2 μg/g)	30 μg/day (3 μg/g)	30 μg/day (3 μg/g)	

^1^ For food supplements exclusively or mainly consisting of dried seaweed, products derived from seaweed, or dried bivalve mollusks. ^2^ The safety assessor must decide whether the trace level of the prohibited substances is toxicologically acceptable and whether the product is safe. ^3^ In the case of using preservatives, such as thiomersal or phenylmercuric salts. ^4^ Concentration in mass percent (weight by weight). ^5^ Established PDE and recalculated permitted concentrations of elemental impurities in drug products, drug substances, and excipients with daily doses of not more than 10 g/day are given in the parentheses.

**Table 2 plants-12-03150-t002:** C-PC samples for heavy metal determination.

Sample Code	Substance	Source	Purification Method	Year
S1	C-PC, lyophilized powder	Cyanobacterial biomass, Kaunas Lagoon, Lithuania (*A. flos-aquae*)	Ion exchange chromatography	2019
S2	C-PC, lyophilized powder	Cyanobacterial biomass, Kaunas Lagoon, Lithuania (*Microcystis* spp.)	Gel filtration chromatography	2020
S3	C-PC, lyophilized powder	Cyanobacterial biomass, Kaunas Lagoon, Lithuania (*Microcystis* spp.)	Gel filtration chromatography	2022
S4	C-PC, lyophilized powder	Cultivated *S. platensis*, (Sigma-Aldrich)	Unknown	Unknown

**Table 3 plants-12-03150-t003:** Instrumental operating conditions of the ICP-MS system.

ICP-MS Parameter	Operation Conditions
Spray chamber	Quartz cyclonic
Sample introduction	MEINHARD^®^ nebulizer
RF ^1^ power	1300 W
Plasma Ar ^2^ flow	18 L/min
Nebulizer Ar flow	1.02 L/min
Auxiliary Ar low	1.20 L/min
Mode	STD
Helium flow for KED ^3^	5.3 mL/min
Dwell time	50 ms
Replicates	3

^1^ Radio frequency, ^2^ argon gas, and ^3^ kinetic energy discrimination.

**Table 4 plants-12-03150-t004:** Data of the limits of detection (LOD) and limits of quantification (LOQ) for lead, cadmium, arsenic, and mercury.

Heavy Metal	LOD (µg/L)	LOQ (µg/L)
Pb	0.002	0.008
Cd	0.002	0.006
As	0.001	0.004
Hg	0.005	0.016

## Data Availability

The data supporting the findings of this study are available from the corresponding author [D.G.] upon reasonable request.

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
