# Peer review of "Determination of Heavy Metal Content: Arsenic, Cadmium, Mercury, and Lead in Cyano-Phycocyanin Isolated from the Cyanobacterial Biomass"

_plants, 2023, doi:10.3390/plants12173150_

Round 1
Reviewer 1 Report
This article measured and compared the content of four heavy metal ions in phycocyanin extracted from different cyanobacteria biomass, and concluded that environmental conditions, algae species, and extraction and purification methods are important factors affecting the content of heavy metal ions in phycocyanin.Overall, although the results of this study have a certain driving effect on understanding the safety of phycocyanin, there are serious shortcomings in writing, data presentation, and other aspects that require extensive and in-depth modifications.
1. The abstract section should be a summary of the research purpose, results, and significance, rather than just a pile of experimental results. This section must be rewritten. In addition, it is best not to include references in the abstract.
2. The keywords section. There are too many keywords, 5 are a suitable number.
3. The introduction section does not provide sufficient background information and the writing logic is not smooth. It is recommended to rewrite it. In this section, it is necessary to inform readers of the background, current research status, and problems faced in this study, and based on this, introduce why this study should be conducted.
4. The results section and discussion section must be reorganized. In this article, the data that should be presented in the results section is placed in the discussion section. In fact, all data should be presented and described in the results section, and experimental data should not appear in the discussion section.
5. There are also areas that require special attention in the presentation of data in this paper. Firstly, the images in this article need to be redone using professional software to improve aesthetics and presentation effects. Secondly, statistical differences between data need to be indicated in the form of letters or symbols in the image. Third, Error bar are necessary for the presentation of results.In the main text, the presentation of data should be mean value ± standard deviation, rather than the current form of mean value (standard deviation).
The language of this paper requires extensive and in-depth editing
Author Response
Dear Reviewer,
Let us express our gratitude for your time and valuable comments on our manuscript. It will help to improve the representation of our experimental data. Please find our response to your comments attached.
Best regards,
Daiva Galinytė

Reviewer 2 Report
Journal: Plants (ISSN 2223-7747)
Manuscript ID: plants-2452761
Type: Article
Title: Determination of the heavy metals: arsenic (As), cadmium (Cd), mercury (Hg), and lead (Pb) content in C-phycocyanin isolated from the cyanobacterial biomass
Section: Phytochemistry
Special Issue: Phytochemical Profiles of Plant Materials: From Extracts to Added Value Ingredients
This manuscript aimed to study the concentrations of the most toxic heavy metals: lead (Pb), cadmium (Cd), arsenic (As), and mercury (Hg) in cyano-phycocyanin (CPC) isolated from cyanobacterial biomass. However, the presentation and writing are not good. I have the following suggestions and comments for the authors to improve the quality of manuscript.
1. Title
Please change “C-phycocyanin” to “cyano-phycocyanin”.
2. Abstract
Lines 28-29
“The extraction and purification of C-PC were carried out according to Khazi et al. with some modifications [1].”
Please delete the reference citation and move it to other sections, e.g. section “4. Materials and Methods”. In general, there is no reference in the abstract.
3. Lines 31-41
“The order of metal concentrations in C-PC isolated from cyanobacterial biomass with dominant genus Aphanizomenon, vary in the range: Pb 0.378 (0.013) µg/g > As 0.150 (0.012) µg/g > Hg 0.050 (0.006) µg/g > Cd 0.020 (0.000) µg/g. Heavy metal concentrations in C-PC isolated from cyanobacterial biomass with dominant genus Microcystis, 2020 range from 0.725 (0.006) µg/g (Pb) followed by 0.443 (0.007) µg/g (Hg), 0.233 (0.005) ( µg/g (Cd) and 0.190 (0.008) µg/g (As). While the concentration of heavy metals in C-PC from cyanobacterial biomass also with the dominant genus Microcystis, but collected in 2022 vary in the range: Pb 0.643 (0.015) µg/g >Cd 0.410 (0.014) µg/g >As 0.160 (0.008) µg/g >Hg 0.021 (0.003) µg/g. The concentration of heavy metals in C-PC from the cultivated Spirulina platensis (S4) was also detected and ranges in the following order: Hg 0.376 (0.016) µg/g >Cd 0.348 (0.005) µg/g >Pb 0.200 (0.008) µg/g > As 0.045 (0.006) µg/g.”
“Pb 0.378 (0.013) µg/g”, “0.725 (0.006) µg/g (Pb)”, the format is not consistent. Please uniform the expressions. Please also check the entire manuscript.
4. Line 58
Please change the citations “[6, 2, 7]” to “[2, 6, 7]”.
5. Lines 41-43
“Environmental conditions, the dominant species of cyanobacteria from which C-PC is extracted, the extraction and purification methods, and the purity grade of C-PC are factors that can influence the content of heavy metals in C-PC.”
How did you make the conclusion that environmental conditions, the extraction and purification methods, and the purity grade of C-PC can influence the content of heavy metals in C-PC? The results from the abstract did not talk about these factors.
6. Section “1. Introduction”
Cyano-phycocyanin (CPC) is extremely valuable in medicine and pharmaceuticals due to its antioxidant, anti-inflammatory, wound-healing, antimicrobial, and anti-cancer activities. Thus, extraction and use of CPC from blooms of cyanobacteria can make use of “waste” biomass when Microcystis scums and blooms are harvested. However, CPC extracted from blooms of cyanobacteria may contain heavy metals, microcystins (MCs) and other cyanotoxins, and other toxic pollutants, which limit their use. In fact, the CPC belong to Added Value Ingredients, and use of CPC from cyanobacteria also fit the scope of the Special Issue “Phytochemical Profiles of Plant Materials: From Extracts to Added Value Ingredients” in the Section Phytochemistry, you submitted to the journal Plants (ISSN 2223-7747). Please insert this interesting point in the section of introduction and cite the following paper.
Chen et al., 2021. Challenges of using blooms of Microcystis spp. in animal feeds: A comprehensive review of nutritional, toxicological and microbial health evaluation. https://doi.org/10.1016/j.scitotenv.2020.142319
7. Lines 55-57
“Metals are non-degradable, remain in the environment for a long time, and exceeding a particular concentration can threaten all living organisms [5].”
Please read and cite the following paper.
Xia et al. 2019. Spatial and interspecies differences in concentrations of eight trace elements in wild freshwater fishes at different trophic levels from middle and eastern China. https://doi.org/10.1016/j.scitotenv.2019.03.134
8. Lines 63-67
“Like many other algae, cyanobacteria have tremendous capability to sorb metal ions. Algae are an essential part of the aquatic ecosystem. They carry out photosynthesis and thus participate in the reoxygenation of water. Studies revealed that specific concentrations of heavy metals affect algal growth, bioactivity, and the process of photosynthesis.”
Please insert the references.
9. Section “4. Materials and Methods”
Lines 463-470
“Gel filtration chromatography on Sephadex G-25 was used to purify C-PC from biomass with dominant Microcystis spp. The column was pre-equilibrated and eluted with 10 mM Na-phosphate buffer (pH 7.0), with the flow rate set at 0.5 mL/min. Purification of C-PC from biomass with dominant Aphanizomenon flos-aquae was carried out by ion exchange chromatography on Q-Sepharose XL. After pre-equilibration of the column with 10 mM Na-phosphate buffer (pH 7.0), elution was performed by linear gradient with 0.5 M NaCl and the same buffer at a flow rate of 0.5 mL/min.”
Why did you use different methods to purify C-PC from biomass with dominant Microcystis spp. or Aphanizomenon flos-aquae?
10. Lines 473-474
“C-PC partially purified from Spirulina platensis was purchased from Sigma- Aldrich (Missouri, USA).”
Please insert the methods of C-PC purification from Spirulina platensis in the revised manuscript.
11. Table 2
Please insert a column to present methods of C-PC purification for different C-PC samples.
12. Section “4.5. Inductively coupled plasma- mass spectrometry”
Please present data of external calibration curves for each element quantified. Please present data of limits of detections (LOD) and limit of quantification (LOQ) for the 4 trace elements. Did you use standard reference material to check the methods for ICP-MS?
13. Section “4.6. Statistical analysis”
Lines 515-518
“The data were statistically calculated and analysed using SPSS V29.0 Metals concentrations were expressed as means, and statistical significance was tested to a significance level of p<0.05 using one-way ANOVA Test.”
Which post hoc test did you use following one-way ANOVA Test? Please insert the information in the revised manuscript.
14. Figure 1
Please also present data of standard deviation by error bars in the figure. Please refer to Fig. 5.
15. Figure 1 caption
What is the number of replicates for analysis? Please insert the information in the revised manuscript.
16. Figure 1
Please present data of statistical analyses. Please use different letters to show significant differences among the different metals. For example, "a" and "b" or "bc" or "c" have significant differences, but "b" and "ab" or "bc" have no significant differences. Then readers can better understand your data.
17. Lines 92-93
“C-PC purity coefficient (A620/A280) from sample S1 was 2.207 (0.015).”
Figure 5
Please cite Fig. 5 here, which should be changed to Fig. 2.
What is the number of replicates for analysis in this figure? What are the error bars, standard deviation (SD) or standard error (SE)? Please insert the information in the revised Fig. 2 caption.
18. Lines 92-93
“C-PC purity coefficient (A620/A280) from sample S1 was 2.207 (0.015).”
What is the relationship between purity coefficient and purity in percentage (% of C-PC in total proteins)? Please also present data of purity of C-PC in percentage.
19. Fig. 2
Similar to Fig. 5, please present the error bars, standard deviation (SD) or standard error (SE). What is the number of replicates for analysis in this figure? Please insert the information in the revised figure caption. Please also present data of statistical analyses by different letters.
20. Figure 3
Please change the order of data from 2013 to 2022. The data of earlier years, e.g. 2013, should be present first (left).
21. Fig. 3
Some data are missing, but it may cause misunderstandings that the values are 0. Please insert text “no data” in the figure, so that readers can better understand your data.
22. Fig. 3, 4
Similar to Fig. 5, please present the error bars, standard deviation (SD) or standard error (SE). What is the number of replicates for analysis in this figure? Please insert the information in the revised figure caption.
23. Table 1
Maximum levels for heavy metals
The units are mg/kg or µg/kg. The units are for wet weight (ww) of dry weight (dw)? Please change the units to mg/kg dw, mg/kg ww, µg/kg dw or µg/kg ww in the revised manuscript.
24. Table 1
For Medical devices, what do the value mean? For example, 0.1 % by mass (w/w). Please make it clear in the revised manuscript.
25. Table 1
For oral, what do the value mean? For example, 5 µg/day (0.5 µg/g). Please make it clear in the revised manuscript.
26. How to evaluate safety (risk of heavy metals) of C-phycocyanin isolated from the cyanobacterial biomass? Please add the analysis. The quantitative analysis of risk will improve quality of this manuscript.
Moderate editing
Author Response

(The authors gave the same response as above.)

Reviewer 3 Report
The Authors investigated the content of heavy metals in phycocyanin samples isolated from natural strains and compared these results with the analysis of phycocyanin purchased from Sigma.
The question remains unclear, why is such a control chosen? Why was the "Sigma" chosen?
In the abstract, in that case, it should be written that the comparison of natural samples was carried out with a commercial sample. The authors write that the comparison was carried out with phycocyanin isolated from cyanobacteria Spirulina grown in the laboratory. The reader wants to know how this strain was grown and suddenly finds out that the authors simply bought a ready-made C-phycocyanin and did not give any characteristics of the strain, cultivation conditions, method of this phycocyanin isolation.
How can the authors explain that cyanobacteria Spirulina cultivated in the laboratory contains heavy metals? For cultivation in the laboratory, a growth medium with a strictly controlled composition prepared with distilled water is used.
There are more comments below.
1) Line 24 Cyanobacteria ( not cyano-bacteria). Please, check also: cyanobacteria or cyanobacterial ?
2) For methodological details description there is a Section "Materials and Methods". Do not place experimental details in the Abstract.
3) Why the Authors did not used Microcystis spp. and/ or Aphanizomenon spp as a control in the laboratory?
4) What recommendations can the authors give about the use of the studied C-PC samples ? How safe is it?
5) What scientific conclusions can be drawn from this work?
6) An article in the journal "Plants" requires a deeper dive into biology of plants or cyanobacteria. In the proposed manuscript, almost no attention is paid to the peculiarities of the biology of the studied cyanobacterial strains and the mechanisms of accumulation of heavy metals by these strains.
This manuscript may be more interesting for a journal dedicated to the purity of water.
7) Please check and correct the English language. Ask a native English-speaking colleague to check the text.
Moderate editing of English language required
Author Response

(The authors gave the same response as above.)

Round 2
Reviewer 1 Report
The paper has made significant revisions and I believe it meets the publication requirements of Plants
Author Response
Dear Reviewer,
Let us express our gratitude again for such a thouhtful review. It really allows us to achieve a better representativeness of the results.
Best regards,
Daiva Galinytė
Reviewer 2 Report
Journal: Plants (ISSN 2223-7747)
Manuscript ID: plants-2452761-peer-review-v2
Type: Article
Title: Determination of Heavy Metal Content: Arsenic (As), Cadmium (Cd), Mercury (Hg), and Lead (Pb) in Cyano-phycocyanin isolated from the Cyanobacterial Biomass
Section: Phytochemistry
Special Issue: Phytochemical Profiles of Plant Materials: From Extracts to Added Value Ingredients
This manuscript aimed to study the concentrations of the most toxic heavy metals: lead (Pb), cadmium (Cd), arsenic (As), and mercury (Hg) in cyano-phycocyanin (CPC) isolated from cyanobacterial biomass.
The revised manuscript improved a lot during the revisions. I have the following comments and suggestions for the authors to improve the quality of the manuscript.
1. Section “1. Introduction”
Lines 45-47
“We aim to isolate C-PC from cyanobacterial biomass collected in nature and employ it in the production of pharmaceutical or cosmetic products.”
Please rephrase this sentence and do not begin with “We”.
2. Figures 1, 3abcd
The presentation of statistical analyses is wrong, e.g., “aa”, “bb”. Please refer to Fig. 2. Please use different letters to show significant differences among the different metals. For example, "a" and "b" or "bc" or "c" have significant differences, but "b" and "ab" or "bc" have no significant differences. Then readers can better understand your data.
3. Please carefully check the entire manuscript, tables, figures and supplementary materials. It is the authors’ responsibility to present their best work to the readers.
minor edits
Author Response
Dear Reviewer,
Let as express our gratitude again for such a thoughtful review. It really allows us to achieve a better representativeness of the results. Please find our response to your comments attached.
Best regards,
Daiva Galinytė

Reviewer 3 Report
The authors answered the reviewer's questions and made the necessary changes to the text of the manuscript.
Author Response
Dear Reviewer,
Let us express our gratitude again for such a thoughtful review. It really allows us to achieve a better representativeness of the results.
Best regards,
Daiva Galinytė
Round 3
Reviewer 2 Report
Journal: Plants (ISSN 2223-7747)
Manuscript ID: plants-2452761-peer-review-v3
Type: Article
Title: Determination of Heavy Metal Content: Arsenic (As), Cadmium (Cd), Mercury (Hg), and Lead (Pb) in Cyano-phycocyanin isolated from the Cyanobacterial Biomass
Section: Phytochemistry
Special Issue: Phytochemical Profiles of Plant Materials: From Extracts to Added Value Ingredients
This manuscript aimed to study the concentrations of the most toxic heavy metals: lead (Pb), cadmium (Cd), arsenic (As), and mercury (Hg) in cyano-phycocyanin (CPC) isolated from cyanobacterial biomass.
The revised manuscript improved a lot during the revisions. I have the following comments and suggestions for the authors to improve the quality of the manuscript.
1. Figures 1
The presentation of statistical analyses is wrong. Please refer to Fig. 2. Please use different letters to show significant differences among the different metals. For example, "a" and "b" or "bc" or "c" have significant differences, but "b" and "ab" or "bc" have no significant differences. Then readers can better understand your data.
Please divide this figure to 4 small figures, similar to Fig. 3 (Each small figure of Fig. 3 presents concentrations of a metal from 4 samples). Each small figure of Fig. 1 can present concentrations of 4 metals from a sample. For example, Fig. 1a can present concentrations of 4 metals of S1. Fig. 1b can present concentrations of 4 metals of S2.
You only need to do statistical analysis of concentrations of 4 metals from each sample. A small figure is independent from others.
2. Fig. 3
The presentation of statistical analyses of Pb, Cd, As and Hg is wrong. Please refer to Fig. 2. Please use different letters to show significant differences among the different metals. For example, "a" and "b" or "bc" or "c" have significant differences, but "b" and "ab" or "bc" have no significant differences. Always indicate the largest value with “a”. A small figure is independent from others.
Then readers can better understand your data. Please refer to Fig. 2 and Fig. 4 of the following paper (Ref. 7 in the list you cited).
Xia W.; Chen L.; Deng X.; Liang G.; Giesy J.P.; Rao Q.; Wen Z.; Wu Y.; Chen J.; Xie P. Spatial and interspecies differences in concentrations of eight trace elements in wild freshwater fishes at different trophic levels from middle and eastern China. Science of the Total Environment 2019, 672, 883-892.
minor edits
Author Response
Dear Reviewer,
Thank you once more for your valuable suggestions and please find our corrections attached.
Best regards,
Daiva Galinytė
